# Analysis of the Mental and Physical Health Symptomatology Scale in a Sample of Emerging and Migrant Adults in Chile

**DOI:** 10.3390/ijerph20064684

**Published:** 2023-03-07

**Authors:** Ana Barrera-Herrera, María José Rivera Baeza, Camila Salazar-Fernández, Diego Manríquez-Robles

**Affiliations:** 1Departamento de Psicología, Facultad de Ciencias de la Salud, Universidad Católica de Temuco, Temuco 4813302, Chile; abarrera@uct.cl (A.B.-H.); maria.baeza@uct.cl (M.J.R.B.); dmanriquezrobles@gmail.com (D.M.-R.); 2Millennium Nucleus to Improve the Mental Health of Adolescents and Youths, Imhay, Santiago 8380455, Chile; 3Laboratorio de Interacciones, Cultura y Salud, Temuco 4813302, Chile; 4Departamento de Análisis de Datos, Facultad de Ciencias Sociales y Humanidades, Universidad Autónoma de Chile, Temuco 4810101, Chile

**Keywords:** health, mental health, measure, emerging adulthood, migrants

## Abstract

Health inequities exist in groups of greater psychosocial vulnerability such as emerging adults and migrants. The study aimed to generate evidence of the validity of the mental and physical health symptomatology scale in two samples of vulnerable groups: emerging university adults, who report high levels of mental health problems, and migrants, who report high levels of physical and mental health problems. Using non-probability sampling, in the first study, evidence of construct validity of the scale was reported in 652 emerging adults and, in the second, evidence of validity was provided from associations with the depression, stress and anxiety scale (DASS-21) among 283 migrants. The results indicate that in Study 1 the two-factor model had adequate indicators of fit and adequate reliability; only the mental health factor presented evidence of convergent validity. In Study 2, the mental health factor showed positive and large associations with the DASS-21, which decreased when the physical health symptoms factor was considered. These analyses provide evidence of validity for the scale, which is an easy-to-use instrument that allows for the assessment of health from an integral perspective.

## 1. Introduction

The World Health Organization understands health as a state of complete physical, psychological, and social well-being that, being in harmony with the environment, exceeds the mere absence of disease [1]. This conceptualisation is dynamic and multifactorial and involves an interaction between its different dimensions [2]. However, despite efforts to promote health from an integral perspective for all people, this continues to be a priority objective due to the increase in inequalities and inequities in vulnerable populations, such as groups living in poverty, women, ethnic minorities, the elderly, emerging adults and international migrants, among other segments of the population [1,3].

One of the main health gaps in the region of the Americas is mental health, a concept that refers to the state of well-being that allows individuals to recognise and make use of their abilities, face the common stresses of life, work productively and thus contribute to the community in which they live [4,5]. Worldwide, mental health problems have increased considerably in recent years, reaching figures that exceed 970 million people [6]. In this regard, the scientific literature has reported that the highest prevalence at the international level is associated with anxious and depressive symptomatology (i.e., 301 million and 280 million, respectively), which in turn has high comorbidities with stress symptomatology. In the Americas region, mental and neurological disorders and suicide represent the second leading cause of years lost due to disability and disability-adjusted life years [4]. In Chile, this trend is consistent with the epidemiological profile yielded by the latest National Health Survey 2016–2017, where a higher prevalence of depression, bipolarity and suicidal ideation and a low percentage of people accessing mental health treatment are observed. Thus, mental health problems constitute the main source of disease burden in the country [7,8].

Physical health refers to bodily well-being and optimal functioning of the organism. Physical symptoms refer to a subjective manifestation of a pathology, appreciable only by the person (e.g., pain, itching, nausea and tiredness), which may or may not be part of a disease [9]; physical symptoms have been widely characterised in health research, being reported by some studies as psychosomatic symptoms, i.e., discomforts frequently linked to mental health alterations (e.g., sleeping, gastrointestinal, sexual and eating problems, and headaches) [10]. In Chile, the latest National Health Survey reports migraine, headaches and insomnia as the most prevalent symptoms in the context of a high existence of risk factors for non-communicable diseases [11]. Despite their scarce study, it is important to consider that physical symptoms have a significant impact on people’s health and functioning, especially those who face adverse environmental situations and psychosocial vulnerability [10,12,13].

### 1.1. Vulnerable Groups: Emerging Adults and Migrants

According to data from the X National Youth Survey of Chile [14], more than four million young people are going through the stage of life called emerging adulthood, a critical period of development between 18 and 29 years of age, where the most characteristic feature is that young people do not perceive themselves as sufficiently adult, postponing milestones associated with adulthood, such as marriage, emotional and financial independence, and having children [15,16]. It is a stage of identity exploration; great self-centeredness; optimism; possibilities; and, at the same time, great instability. It is also a crucial period because young people decide on a life project linked to work and/or higher education [16,17,18]. At this stage of life, especially for young university students, various psychosocial stressors related to the challenges of emerging adulthood converge, in addition to the characteristics of the university environment, which can lead to the appearance of symptoms or various mental health disorders [19,20,21,22].

Both nationally and internationally, the prevalence of mental health symptoms and disorders such as depression, anxiety, stress, adaptive disorders, eating disorders, and suicidal risk is high among university students [11,21,22,23,24]. These data on mental health symptomatology contrast with those on physical health since emerging adults are characterised by good physical health, with a lower prevalence of chronic diseases and a low risk of death [15,25]. However, national studies show that university students exhibit physical risk behaviours associated with health habits, such as diet, sleep patterns and substance consumption [19].

Migration has become a sociodemographic phenomenon of worldwide relevance. The migrant population settled in Chile amounts to more than 1.6 million people, which represents 9% of the national population [26]. It is important to consider that the decision to migrate has various motivations, both those related to the search for greater welfare, family reunification, and better labour and economic opportunities and those derived from socio-political conflicts in the countries of origin [27,28]. Scientific evidence has corroborated that this process presents various difficulties that extend from the pre-migration phase and the migration journey itself to the arrival and settlement in the host country [29]. Research has linked migration processes to symptoms of stress, anxiety, depression, psychosomatic symptoms (e.g., headaches and stomach problems) and mental health disorders in general [30,31,32,33,34,35,36,37,38,39,40].

### 1.2. Instruments to Assess the State of Mental and Physical Health

A large number of instruments evaluate symptomatologies of the most prevalent mental health conditions for university emerging adults, such as anxiety, depression, stress, substance use, eating disorders, and suicidal ideation or risk. Locally, one of the most widely used is the depression, anxiety and stress scales, DASS-21 [41], followed by scales to evaluate specific symptomatologies such as the patient health questionnaire (PHQ-9) [42], Beck depression inventory (BDI-II) [43,44], Goldberg health questionnaire (GHQ-12) [45,46] and Beck anxiety inventory (BAI) [44,47,48]. Other investigations have used clinical interviews, considering the diagnostic criteria of the DSM IV TR [21,49,50] (see Table 1). When analysing these instruments for university emerging adults, some difficulties arise: although several scales assess the same mental health problems, the prevalence reported in each research is usually different since some studies analyse mental health symptomatology while others report the prevalence of clinical disorders. This implies the use of different instruments, such as screening scales or diagnostic interviews [19]. Furthermore, although some instruments claim to assess general health status, they give priority to the most prevalent mental health conditions, with less attention being given to the assessment of physical health status or health habits.

Concerning the migrant population, particularly in Chile, mental health symptomatology has been evaluated using the clinical symptomatology inventory SCL-90 to evaluate psychological distress [51], the WHOQoL-Bref questionnaire to estimate the quality of life [52], the OQ questionnaire to assess emotional adjustment [32], the cultural formulation interview included in DSM 5 to improve diagnosis and care in migrant users by estimating the extent to which migration entails emotional consequences [53], and the positive and negative affect scale [54]. A detailed description of these instruments can be found in Table 1. After revising the most commonly used instruments for assessing mental health, evidence indicated that the DASS-21 scale demonstrated good evidence of validity and standardization, along with adequate psychometric properties [55]. This allows for the screening of the population with clinical symptoms that require professional support. Additionally, it has been widely used in both Chilean and Hispanoamerican samples.

Most of these instruments do not directly estimate mental health indicators related to depression, anxiety and stress, nor do they provide joint information that would allow for effective discrimination between these symptomatologies. In addition, few instruments report evidence of psychometric properties, being for the migrant population a source of difficulties related to the screening, diagnosis and treatment of mental health diseases, directly affecting the health and quality of life of this population group [52,56].

As can be seen, emerging adults and migrants are characterised by facing significant public health challenges since, unlike the general population, they are constantly exposed to the stressors and demands of the context, which, when they exceed the available resources of the subjects, have been associated with mental and physical health symptoms that need to be evaluated. Although instruments are available to assess the mental health status of both population groups, difficulties occur related to the variety of screening scales to assess the same disorders, the lack of evidence of validity in some scales, a poor ability to discriminate between different disorders and a lack of inclusion of indicators associated with the measurement of physical health. In addition, some instruments are lengthy and have complex instructions. Taking into account this scenario, a more precise, direct and easy-to-apply measure is necessary for these at-risk groups, an instrument that can jointly assess both mental and physical health symptomatology by considering the interrelation of such symptomatologies. Therefore, this study aims to generate evidence of the validity of a scale that considers this integral evaluation of health (mental and physical health symptomatology scale) in one sample of university emerging adults and another of migrants in southern Chile. The specific objectives are (a) to obtain evidence of the construct validity of the health symptomatology scale in a sample of university emerging adults and (b) to generate evidence of the validity of the mental and physical health symptomatology scale from its association with other mental health variables. The evidence presented above allows us to hypothesise that the constructs of mental health symptomatology and physical health symptomatology, although related, should be differentiated, especially when dealing with vulnerable samples. The hypothesis proposes that health symptomatology comprises two separated factors, one relating to mental health symptoms and the other to physical health symptoms. These factors should be correlated because they measure the same construct but independent of each other as they measure different aspects of the health symptomatology. Then, based on the associations with a scale that evaluates depression, anxiety and stress (DASS-21), we expect to find positive associations of moderate magnitude with the mental health symptomatology factor.

## 2. Study 1

This study aimed to provide evidence of the construct validity of the mental and physical health symptomatology scale in a sample of emerging adults obtained from the Fondecyt Initiation Project N° 11,200,984 “Positive development, mental health and psychosocial variables involved: Towards a comprehensive model in Chilean university students”.

### 2.1. Participants

Through a non-probabilistic convenience sampling, 652 university students from two regional universities participated. The sample size was estimated using Soper’s [57] a-priori sample size calculator for structural equations models. This calculation considered an anticipated effect size of 0.2, a desired statistical power level of 0.8, a probability level of 0.05, and the number of the observed and latent variables in the main project frow which we obtained this data. The inclusion criteria considered that the participants were university students belonging to the region of La Araucanía. The exclusion criteria considered people with visual or hearing impairments since the instrument is not adapted for this type of case. The age range was 18 to 29 years (*M* = 21, *SD* = 2.39). Concerning gender, 65.3% reported belonging to the female gender, 32.3% to the male gender and 2.3% to the non-binary gender. Regarding marital status, 99.2% of the students were single, 58% of the sample belonged to the middle socioeconomic level, 43.2% of the students lived independently from their parents and 23.5% worked in addition to studying (Table 2).

### 2.2. Measures

The participants answered a self-report questionnaire containing sociodemographic variables and the scale of study in this research.

Mental and physical health symptomatology scale (MPHS-Scale) [31]. This scale is composed of eight items that assess the frequency of health symptomatology. Specifically, individuals were asked to report how often they have felt a list of symptoms (e.g., anxiety, depression, stress, sleeping problems, eating problems, stomach problems, sexual problems and headaches) using a 7-point Likert-type scale (1 = never to 7 = very often). Higher scores reflected a greater presence of this type of symptomatology. A previous study [31] revealed the presence of two factors, one grouping the first three items in terms of mental health symptoms and the rest in terms of physical symptoms. It also showed that the correlation between the mental health symptoms factor and physical health symptoms was high. In this study, the reliability of the mental health and physical factors was good, with alpha coefficients of 0.746 and 0.701, respectively.

### 2.3. Procedures

Participants completed an online survey through the Survey Monkey platform, which included informed consent approved by the Ethics Committee of the authors’ university of affiliation (Res. N° 48/20). Through the authorisation of two universities in southern Chile, we proceeded to disseminate the study through physical posters and social networks. Participants responded to the survey in approximately 20 min. The data were collected between September and October 2022; after reaching the expected sample, 20 bookstore gift cards were raffled among the study participants as a form of compensation.

### 2.4. Analysis

First, the descriptive and frequency statistics of the MPHS-Scale were reviewed. Of the total 652 participants, 647 provided complete data for scale analysis. Then, whether the data complied with the multivariate normality assumption was analysed, and it was found that the data did not comply with this assumption (*p* < 0.05). In addition, one-sample t-tests were performed to assess whether the item scores were significantly greater than the midpoint of the scale. A significance level of 0.05 was used. Since this scale had antecedents that proposed a two-factor structure conceptualised as mental health symptoms and physical health symptoms, and a confirmatory factor analysis (CFA) was performed on the data. To estimate the model, we used the robust estimation of diagonalised weighted least squares (DWLS), which is appropriate when the data do not meet the assumption of normality and are intervalar [58]. To evaluate the model fit, we used different indicators such as the comparative fit index (CFI), the Tucker Lewis index (TLI), the standardised root mean squared residuals (SRMR), and the root mean squared error of approximation (RMSEA) with its 90% confidence interval. These indicators were interpreted according to conventional goodness-of-fit criteria: CFI and TLI > 0.95 and SRMR and RMSEA ≤ 0.08. To evaluate the internal consistency of the factors, composite reliability (CR) was used. If CR > 0.70, it is an indicator of good reliability. Then, evidence of convergent validity was evaluated through the average variance extracted (AVE). The interpretation of this indicator holds that AVE values > 0.50 and lower than the CR value indicate that the variance explained by the construct is greater than the measurement error and greater than the factor loadings, obtaining evidence of convergent validity [59]. Finally, reliability was evaluated via Cronbach’s alpha, whose values above 0.70 are considered acceptable [60,61].

### 2.5. Results

The descriptive statistics of the variables of the mental and physical health symptomatology scale are shown in Table 3. This table shows that the emerging adult participants showed scores significantly higher than the midpoint of the scale (score equal to four) in the anxiety and stress items, while they presented scores significantly lower than the midpoint of the scale in depression, eating, stomach and sexual problems, and headache. No statistically significant differences were found with the midpoint of the scale in sleeping problems.

Regarding the evaluation of the scale structure, a two-factor model was specified in accordance with previous results [62]. This two-factor structure reported an excellent fit in the sample of emerging adults, *χ2*(19) = 49.068, *p* < 0.05, CFI = 0.987, TLI = 0.980, RMSEA = 0.049 [0.033, 0.067], and SRMR = 0.047. The factor loadings of this model ranged from 0.355 to 0.755. Figure 1 shows the configuration of the model and the factor loadings of the items in each factor. The AVE and CR indicators were 0.54 and 0.77 for the mental health factor and 0.31 and 0.70 for the physical health factor, respectively. Following the interpretation of the composite reliability of both factors was adequate (CR > 0.70), which agrees with Cronbach’s alpha values of 0.75 and 0.70, respectively [61]. As for evidence of convergent validity [59], the criteria were met for the mental health factor (AVE = 0.54 > 0.50 and <CR = 0.78), whereas, in the physical health symptoms factor, although the AVE is higher than the CR, its value was lower than 0.50. This indicates that the mental health factor presented evidence of convergent validity, whereas the physical health factor did not.

To test an alternative model, a single-factor model was specified, which showed an unacceptable fit because most of its indicators were outside the expected ranges, *χ*^2^(20) = 122.467, *p* < 0.05, CFI = 0.922, TLI = 0.901, RMSEA = 0.089 [0.074, 0.104], and SRMR = 0.054. The AVE and CR indicators were 0.40 and 0.83, indicating that the model possesses good internal consistency, also corroborated by Cronbach’s alpha = 0.80; however, it does not meet the criteria to provide evidence of convergent validity. Consequently, this one-factor model was not adequate.

## 3. Study 2

The objective of Study 2 was to provide evidence of the validity of the mental and physical health symptomatology scale from associations with other constructs using the DASS-21 scale in a sample of migrants. For this study, we used data obtained from the Fondecyt initiation project N° 11,181,020 “Acculturation and mental health in migrant population in the region of La Araucanía, the role of mediators and moderators”.

### 3.1. Participants

Through online convenience sampling, 283 migrants participated. The sample size was estimated based on the absolute number of cases criterion. Specifically, the estimation was calculated with a 95% confidence level and a 5% margin of error of the total number of immigrants living in the La Araucanía region. Inclusion criteria considered that participants were migrants over 18 years of age, from Venezuela, Colombia, or Haiti and residing in the Araucanía region, excluding those who were in the area for tourism. The age range of the participants was 18 to 72 years (*M* = 34.5, *SD* = 9.53), 83% from Venezuela, 13% from Colombia, and 4% from Haiti. A further description of the characteristics of the participants can be found in Table 2.

### 3.2. Instruments

For this study, the mental and physical health symptomatology scale [31] and a sociodemographic questionnaire were used to characterise the sample.

Depression, anxiety and stress scales (DASS-21) [63]. The scale is composed of 21 items that evaluate the frequency with which symptoms related to depression, anxiety and stress have been experienced during the last week. The depression subscale measures symptoms such as a lack of pleasant feelings about life, a lack of interest, a lack of self-appreciation, the devaluation of life, and discouragement. The anxiety subscale assesses autonomic nervous system arousal, physiological tension and agitation, situational anxiety, and subjective experiences of anxiety, and the stress subscale assesses difficulty in relaxing, easy agitation, irritability or overreaction, and impatience. It has a four-point Likert-type response format (from 0 = does not describe anything that happened to me or how I felt in the week to 3 = yes, this happened to me a lot, or almost always). The higher the score on the global scale and its dimensions, the greater the presence of symptomatology. This instrument has adequate psychometric properties in Chilean [41,64], Peruvian [65], and immigrant populations [66,67]. Specifically, it has shown a solid internal consistency (Cronbach’s alpha > 0.70 in its three scales) and adequate evidence of construct validity confirming its three factors [41]. Additionally, this instrument has evidence of concurrent and divergent validity, using the Beck depression inventory-II and the Beck anxiety inventory [41,64].

### 3.3. Procedures

The Ethics Committee of the authors’ university of affiliation approved this study (Res. N° 21/18). The survey was administered online via Google Forms from July to November 2021. This format allowed access to participants while reducing the risk of COVID-19 contagion. It included an informed consent form indicating the purpose of the study, anonymity and confidentiality were guaranteed, and the contact details of the responsible investigators were provided. Participation in the study was remunerated with about 10 USD. Responding to the survey took approximately 15 min.

### 3.4. Analysis

As in Study 1, descriptive and frequency statistics were reviewed and the multivariate normality assumption was analysed, with the latter being rejected (*p* < 0.05). Then, one-sample t-tests were performed to assess whether item scores were significantly greater than the midpoint of the scale, using a significance level of 0.05. Due to the results of Study 1, a structural equation model was tested using robust DWLS estimation. Two structural equation models were tested: (1) using the mental and physical health symptomatology scale and assessing its correlations with the DASS-21 depression, anxiety and stress factors and (2) only using the mental health symptomatology scale and its associations with the DASS-21 factors. This second model was tested because the AVE and CR indicators of the mental health symptomatology subscale were adequate, whereas the same was not true for the physical health symptom dimension. To assess the fit of the model, the same fit indicators from Study 1 were used.

### 3.5. Results

The descriptive statistics of the variables of the mental and physical health symptomatology scale for migrants are shown in Table 4. This table shows that the migrant participants showed scores significantly higher than the midpoint of the scale (score equal to four) only in the stress item, while they presented scores significantly lower than the midpoint of the scale in the items of depression, eating, stomach and sexual problems, and headache. No statistically significant differences were found with the midpoint of the scale in the items of anxiety and sleeping problems.

Regarding Model 1, which sought to evaluate the associations between the mental and physical health symptomatology scale with the depression, anxiety and stress dimensions of the DASS-21, it was found that this model presented an adequate fit, *χ*^2^(367) = 460.815, *p* < 0.05, CFI = 0.961, TLI = 0.956, RMSEA = 0.030 [0.020, 0.039], and SRMR = 0.052. The correlations of the mental health subscale with the depression, anxiety and stress factors of the DASS-21 were *r* = 0.497, *p* < 0.05; *r* = 0.176, *p* = 0.117 and *r* = 0.430, *p* < 0.05, respectively. Meanwhile, the correlations between the physical health subscale and the DASS-21 factors were depression: *r* = 0.176, *p* = 0.224; anxiety: *r* = 0.503, *p* < 0.05; and stress: *r* = 0.256, *p* = 0.080.

Next, Model 2, comprising only the mental health symptoms factor and the three scales of the DASS-21, was tested. This model presented an acceptable fit, *χ^2^*(370) = 476.435, *p* < 0.05, CFI = 0.955, TLI = 0.951, RMSEA = 0.032 [0.023, 0.040], and SRMR = 0.053. Correlations between the mental health factor and the DASS-21 factors were high. Specifically, with the depression factor of the DASS-21, the correlation was *r* = 0.665, *p* < 0.05; with the anxiety factor, *r* = 0.667, *p* < 0.05; and with the stress factor, *r* = 0.675, *p* < 0.05. Figure 2 shows, for comparison, the two estimated models.

## 4. Discussion

The present study aimed to provide evidence of the validity of the mental and physical health symptomatology scale in one sample of university emerging adults and another of migrant residents of southern Chile. In the first study, using a sample of emerging adults, we reported evidence of construct validity of this scale, which was composed of a mental health factor and a physical health factor. Then, in a second study, a sample of migrants was used to evaluate the associations of these two factors with a scale widely used to assess mental health, such as the DASS-21. Two models were tested; the first evaluated the associations between the mental health and physical health factors with depression, anxiety and stress factors, and the second the associations between the mental health factor and the three factors of the DASS-21. The results suggest that the mental health factor of the health symptomatology scale shows positive and large associations with the DASS-21, which decreased when considering the physical health symptoms factor [66,68,69].

As reported in mental health studies [21,22,23,24], the emerging adult university students in this sample presented high scores for anxiety and stress symptomatology items. These findings reaffirm the vulnerability of university students to the development of emotional disorders [41]. The comorbidity of these conditions is consistent with the national prevalence, which suggests that conditions such as anxiety and stress are the most frequent mental health problems in emerging adulthood. These conditions have been explained by the perception many young people have of being overwhelmed by demands, due to the multiple psychosocial stressors they must face adapting to their new vital contexts. These include emerging adulthood, the challenge of becoming adults, and the demands of the university environment associated with a more demanding academic environment, which demands greater autonomy and the necessary skills to cope with stressors, such as migrating from their parents’ home, economic instability, and combining studies with parenting and/or work [17,21,22,70].

However, a striking result is that, in this descriptive analysis, no scores significantly different from the midpoint were found for the depression item, although it is one of the most prevalent conditions in this population [19,21,22]. This result could be explained by the wording of the item, which could suggest a depressive disorder and not a symptom of depression. Evidence shows that in screening instruments, university students recognise without difficulty symptoms associated with depression, such as low mood, loss of motivation, sadness and hopelessness; probably identifying with the term “depression” may make some young people think that they should have previously received a professional diagnosis before answering positively to this item, or else they may not declare this symptomatology due to the stigma associated with presenting mental health problems [71].

Regarding the physical symptoms evaluated in the scale, a positive element is that the sample presented scores significantly lower than the midpoint of the scale in eating, stomach, sexual and headache problems. This is consistent with previous research maintaining that most emerging adults have a good state of physical health and a low incidence of problems associated with chronic diseases [15,25]. However, it is necessary to consider that in the group of university students, risk behaviours related to diet, sleep and sexual life can eventually lead to mental health disorders. Therefore, the association between mental health symptoms and physical health symptoms allows us to argue that these should not be neglected despite their underreporting and that measures should be taken to prevent their appearance or comorbidity.

Regarding the structure of the mental and physical health symptomatology scale, it was found that the two-factor model showed adequate fit indicators. However, despite this, only the mental health factor presented adequate evidence of convergent validity, indicating that it is a more robust factor than its pair associated with physical symptomatology. A possible explanation for the low evidence of convergent validity for the physical health factor may be the low prevalence of physical symptoms or diseases at this stage of the life cycle since most of the psychological disorders that affect the adult life of the subjects have their first appearance in adolescence and/or emerging adulthood [70]. Future studies with this same sample should evaluate whether these data are replicated or respond to the specific characteristics of this sample of emerging adults. Despite the above, this first study, in summary, confirms a good performance of the mental and physical health symptomatology scale, and of both factors when evaluating the two symptomatologies in the sample of emerging adults.

On the other hand, according to the literature on migration processes and mental health, migrants presented scores significantly higher than the midpoint for stress symptoms. However, this trend is not observed for depressive and physical symptoms. A possible explanation is the acculturation process to which migrants are subjected as part of the adaptation they must undergo, that is, adapting to a new way of living, customs, speaking, and relating in a country with a bureaucratic system that is not always friendly to new residents. This can lead to increased levels of stress to which immigrants are subjected, omitting depressive or physical symptoms that they may present to achieve the expectations of migration. Another possible explanation is related to the phenomenon of the healthy migrant, i.e., we would be facing a first wave of immigration in which people have better health conditions and more capabilities and resources with which to migrate [72,73,74,75]. This phenomenon has been studied in Chile, and research indicates that this phenomenon tends to disappear in migrants who have resided in the country for more than ten years [76,77].

However, despite the results in this sample, it is relevant to mention that the indicators worldwide are worrisome, and in Chile, they have increased considerably as a result of the socio-political crisis that the country has faced. Despite the economic growth observed, this only benefits an elite of the population, which in turn accumulates discontent and impoverishes those groups that are already in a situation of vulnerability, affecting mental health problems [72,78].

Regarding the scale, Model 2 shows evidence of the validity by association with another measure as support for the use of the mental health factor as a more direct measure of mental health symptomatology, free of bias in the interpretation of symptoms and easier to apply than the DASS-21. However, although the first model showed no association between physical symptoms and depression or between physical symptoms and stress, the correlation was high between physical symptoms and anxiety. The latter may indicate that the presence of both symptoms implies the presence of a high comorbidity. Thus, anxiety was positively associated with sleeping, eating, stomach, sexual and headache problems. In this way, those symptoms that generate a lower presence of conflicts with social desirability were significantly reported.

As can be seen, having brief and easy-to-use instruments such as the mental and physical health symptomatology scale, which can be applied to the entire population, i.e., which are understandable and free of interpretation biases, makes it possible to carry out studies that compare results in samples that at first sight may be diverse but that share certain characteristics of vulnerability.

The present research constitutes a valuable contribution to the understanding of health from an integral perspective, highlighting the importance of mental and physical health, for which instruments with adequate psychometric properties are required. However, limitations should be considered. Both studies developed are cross-sectional; therefore, future studies should consider longitudinal approaches to evaluate the temporal stability or evolution of these results (i.e., test-retest consistency). Another limitation corresponds to the type of sampling used and the characteristics of the samples; future research should look for more heterogeneous samples. In addition, future lines of research should analyse evidence of convergent validity of the mental and physical health symptomatology scale with a measure that assesses physical health symptoms.

## 5. Conclusions

Considering the results, this study is relevant because it provides an instrument with adequate psychometric properties and evidence of construct validity, and, through its association with other mental health variables, it is widely used as is the DASS-21 and adequate reliability. This instrument is easily interpretable, quick to use, free of bias, has few items, and contemplates a general view of health, including the main symptoms of mental and physical health of the population.

## Figures and Tables

**Figure 1 ijerph-20-04684-f001:**
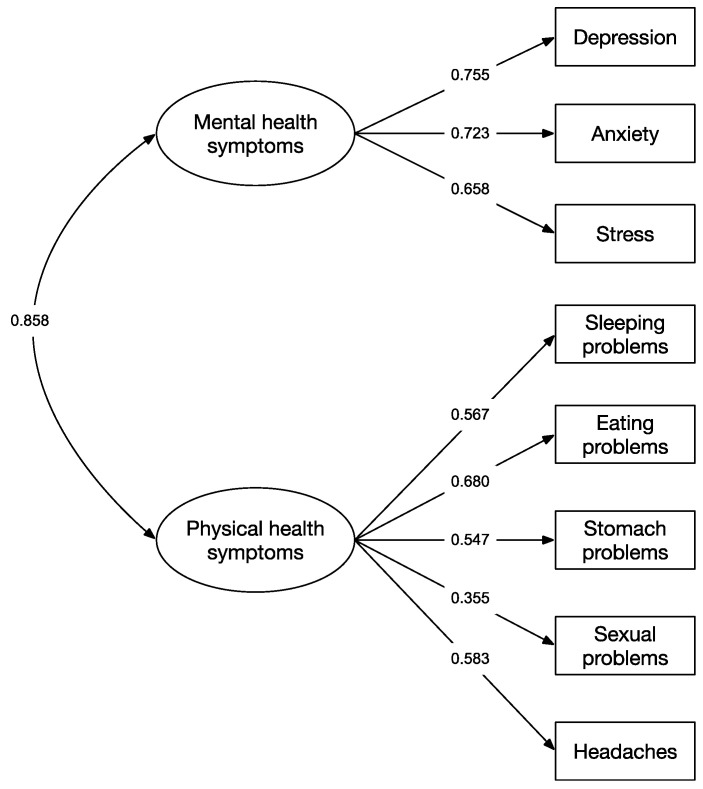
Confirmatory model of the scale of mental and physical health symptoms in the sample of emerging adulthood (*n* = 647).

**Figure 2 ijerph-20-04684-f002:**
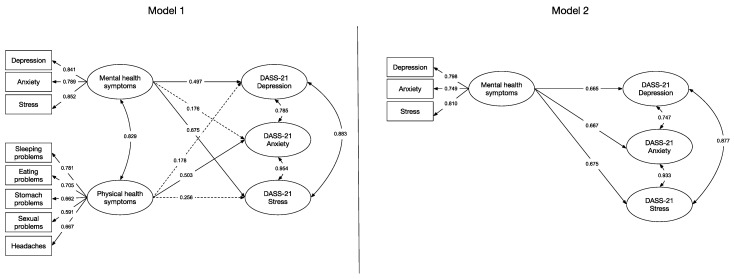
Model comparison of the associations between the scale of mental and physical health symptoms and DASS-21.

**Table 1 ijerph-20-04684-t001:** Instruments to Assess the State of Mental and Physical Health in emerging adults and migrants.

Name	No. Items	Response Format	Factors	Population	Country	Reliability	Validity
Brief scale for the evaluation of acculturation stress in the migrant population (EBEA) [35]	14	Likert from 1 (nothing stressful) to 5 (very stressful).	(1) Preparation and departure from the country of origin, (2) socioeconomic concerns and (3) adaptation to the receiving society.	1309 adults between 18 and 60 years of age, from Colombia and Peru.	Chile	A ω between 0.83 and 0.92 for the subscales.	Concurrent (convergent, discriminant) and construct validity.
Adaptation of the clinical symptomatology inventory SCL-90 [51]	50	Likert from 0 (nothing at all) until 4 (much or very much).	(1) Somatisation, (2) interpersonal awareness, (3) depression, (4) anxiety and (5) hostility.	109 adults between 18 and 76 years of age, from Bolivia, Colombia and Peru.	Chile	Cronbach’s alpha of 0.95.	-
WHOQoL-Bref questionnaire [52]	26	Likert from 1 to 5.	(1) General quality of life, (2) health satisfaction, (3) physical domain, (4) psychological domain, (5) psychological domain and (6) social domain.	431 adults from Colombia and Peru.	Chile	-	-
OQ questionnaire [32]	35	Likert from 0 (never) to 4 (almost always).	(1) Interpersonal relationships, (2) social role and (3) symptoms.	431 adults between 18 and 76 years of age, from Colombia and Peru.	Chile	An α of 0.91 for the scale, with an α between 0.65 and 0.88 for the subscales.	-
Cultural formulation interview (DSM 5) [53]	-	-	-	-	-	-	-
Positive and negative affect scale [54]	12	Likert from 1 (never) to 6 (always).	(1) Positive affect, (2) negative affect.	517 adults between 18 and 76 years of age, from Chile, Colombia, Peru and Venezuela.	Chile	An α of 0.89 for the scale, with an α between 0.75 and 0.92 for the subscales.	Construct and concurrent validity.
Depression, anxiety and stress scale, DASS-[21,41]	21	Likert format, from 0 (does not describe anything that happened to me or how I felt during the week) to 3 (yes, this happened to me a lot or almost always).	(1) Depression, (2) anxiety and (3) stress.	484 undergraduate students, between 18 and 28 years of age.	Chile	Cronbach’s alpha of 0.91, and between 0.73 to 0.85 for the subscales.	Construct, concurrent and divergent validity.
Patient health questionnaire (PHQ-9) [42]	9	Likert format from 0 (never) to 3 (almost every day).	Depression.	1327 adults from the south of Chile.	Chile	Cronbach’s alpha of 0.83.	Construct and concurrent validity.
Inventario de Depresión de Beck (BDI-II) [43,44]	21	Likert scale from 0 to 3.	Depression.	484 undergraduate students, between 18 and 28 years of age.	Chile	Cronbach’s alpha of 0.91.	Discriminant validity.
Goldberg health questionnaire (GHQ-12) [45,46]	12	Likert format from 0 (better than usual) to 3 (much less than usual).	1 to 3 factors, depending on the study.	963 undergraduate students.	Chile	Cronbach’s alpha of 0.86.	Construct validity.
Beck anxiety inventory (BAI) [44,47,48]	21	Likert scale from 0 to 3.	Anxiety.	484 undergraduate students, between 18 and 28 years of age.	Chile	Cronbach’s alpha of 0.91.	Discriminant validity.

**Table 2 ijerph-20-04684-t002:** Sociodemographic descriptions of the sample (emerging adults and international migrants).

	Study 1	Study 2
	n (Percentage/Mean (DS))	n (Percentage/Mean (DS))
Age	21.0 (2.39)	34.5 (9.53)
Country of origin		
Colombia	-	36 (13%)
Venezuela	-	235 (83%)
Haiti	-	12 (4%)
Chile	652 (100%)	-
Marital status		
Single	646 (99.2%)	135 (47%)
Married	2 (0.3%)	69 (24%)
Divorced	-	7 (2%)
Separated	-	8 (3%)
Cohabiting	4 (0.5%)	64 (23%)
Sex		
Female	424 (65.3%)	189 (67%)
Male	215 (32.3%)	94 (33%)
Non-binary	13 (2.3%)	-
Socioeconomic status		
Low, very low	196 (30%)	158 (55%)
Medium	378 (58%)	120 (42%)
High, very high	78 (12%)	5 (3%)
Educational level		
High school completed or lower	-	84 (30%)
University completed or incomplete	652 (100%)	177 (62%)
Postgraduate	-	22 (8%)
Time living in Chile (months)	-	32.12 (22.4)

**Table 3 ijerph-20-04684-t003:** Descriptive statistics and comparisons with the middle point of the scale of mental and physical health symptoms in the sample of emergent adulthoods.

Item	Mean	Standard Deviation	Midpoint Comparison = 4
1. Anxiety	4.96	1.67	*t*(647) = 14.54, *p* < 0.05
2. Stress	5.00	1.57	*t*(647) = 16.19, *p* < 0.05
3. Depression	3.36	1.96	*t*(647) = −8.33, *p* < 0.05
4. Sleeping problems	3.97	2.01	*t*(647) = −0.33, *p* = 0.37
5. Eating problems	3.09	2.00	*t*(647) = −10.15, *p* < 0.05
6. Stomach problems	3.20	2.05	*t*(647) = −8.30, *p* < 0.05
7. Sexual problems	1.96	1.62	*t*(647) = −32.00, *p* < 0.05
8. Headaches	3.73	1.96	*t*(647) = −3.46, *p* < 0.05

**Table 4 ijerph-20-04684-t004:** Descriptive statistics and comparisons with the middle point of the scale of mental and physical health symptoms in the sample of migrants.

Item	Mean	Standard Deviation	Midpoint Comparison = 4
1. Anxiety	4.01	1.98	*t*(282) = 0.120, *p* = 0.91
2. Stress	4.59	1.90	*t*(282) = 5.22, *p* < 0.05
3. Depression	3.48	2.07	*t*(282) = −4.20, *p* < 0.05
4. Sleeping problems	3.87	2.13	*t*(282) = −1.03, *p* = 0.30
5. Eating problems	2.59	1.72	*t*(282) = −13.84, *p* < 0.05
6. Stomach problems	2.85	1.96	*t*(282) = −9.86, *p* < 0.05
7. Sexual problems	2.52	1.99	*t*(282) = −12.57, *p* < 0.05
8. Headaches	3.63	2.07	*t*(282) = −3.01, *p* < 0.05

## Data Availability

Not applicable.

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
