# Peer review of "Analysis of the Mental and Physical Health Symptomatology Scale in a Sample of Emerging and Migrant Adults in Chile"

_ijerph, 2023, doi:10.3390/ijerph20064684_

Round 1

Reviewer 1 Report

The present research constitutes a valuable contribution to the understanding of health from an integral perspective, highlighting the importance of mental and physical health, for which instruments with adequate psychometric properties are required.

Thus, it is hypothesised that the structure of the health symptomatology construct should be composed of two independent but related factors  that allow a differentiated understanding of mental health symptomatology and physical health symptomatology. – The sentence is incomprehensible. Please be more precise in expressing your thoughts.

2.3. Procedures (study 1)

The data were collected between September and October 2022; after reaching the expected sample, 20 bookstore gift cards were raffled among the study participants as a form of compensation.

How the expected sample size was determine?

Author Response

Dear Reviewer:
We appreciate the time and effort you put into providing comprehensive comments on our manuscript. We appreciate your valuable and insightful comments to improve our document.
We have incorporated the suggestions and responded point by point to the observations (Please see attached).
Sincerely,
The authors

Response to Reviewer

Comments: The present research constitutes a valuable contribution to the understanding of health from an integral perspective, highlighting the importance of mental and physical health, for which instruments with adequate psychometric properties are required.

Response: Thank you for your comments and thoughtful review. We appreciate the consideration of our article to revise-and-resubmit.

Comments: Thus, it is hypothesised that the structure of the health symptomatology construct should be composed of two independent but related factors that allow a differentiated understanding of mental health symptomatology and physical health symptomatology. – The sentence is incomprehensible. Please be more precise in expressing your thoughts.

Response: Thank you for this suggestion. In the revised manuscript, we have clarified the hypothesis of our study. On lines 155-159 you can see these changes:

“The hypothesis proposes that health symptomatology is comprised of two separated factors, one relating to mental health symptoms and the other to physical health symptoms. These factors should be correlated because they measure the same construct, but independent of each other as they measure different aspects of the health symptomatology”.

Comments: 2.3. Procedures (study 1)

The data were collected between September and October 2022; after reaching the expected sample, 20 bookstore gift cards were raffled among the study participants as a form of compensation. How the expected sample size was determine?

Response: Thanks for pointing this out. We have included on lines 169 -173 how we estimated sample size:

“The sample size was estimated using Soper’s (2022) a-priori sample size calculator for structural equations models. This calculation considered an anticipated effect size of .2, a desired statistical power level of .8, a probability level of .05 and the number of the observed and latent variables in the main project frow which we obtained this data”.

Reviewer 2 Report

First of all thank you for your contribution, it is very important to analyse the relationship between physical health and mental health symptoms.

1.- In the introduction different measuring instruments are detailed but no justification is given as to why DASS-21 in particular is chosen.

2.- In study 1 the sections are mixed up, in the section on measures it talks about outcomes, please review the sections on measures, procedure and outcomes.

3.- In the mental and physical health scale, the description is poor, please add some more details.

4.- Regarding the sample, why was the final sample size considered in both studies? How was the estimation made?

5.- In the sample of study 1, university students in one age range were considered, however, in study 2 a wide age range was considered, why was the age range widened in study 2?

6.- DASS-21 has been used to correlate with the mental and physical health symptomatology scale, in this case because a scale with dimensions of physical functioning has not been used.

Author Response

Dear Reviewer:
We appreciate the time and effort you put into providing comprehensive comments on our manuscript. We appreciate your valuable and insightful comments to improve our document.
We have incorporated the suggestions and responded point by point to the observations (Please see attached).
Sincerely,
The authors

Response to Reviewer

Comments: First of all thank you for your contribution, it is very important to analyse the relationship between physical health and mental health symptoms.

Response: We are grateful for the valuable comments made on this article.

Comments: 1.-In the introduction different measuring instruments are detailed but no justification is given as to why DASS-21 in particular is chosen.

Response: We greatly appreciate this comment. We have provided evidence to support why we choose the DASS-21 scale on our study. See lines 120-125:

“After revising the most commonly used instruments for assessing mental health, evidence indicated that the DASS-21 scale demonstrated good evidence of validity and standardization, along with adequate psychometric properties (Oei et al., 2013). This allows for screening of the population with clinical symptoms that require professional support. Additionally, it has been widely used in both Chilean and Hispanoamerican samples”.

Comments: 2.- In study 1 the sections are mixed up, in the section on measures it talks about outcomes, please review the sections on measures, procedure and outcomes.

Response: As suggested, we have reviewed the measures section and reorganized it accordingly. Please refer to comment 3 for the specific changes that were made to the manuscript.

Comments: 3.- In the mental and physical health scale, the description is poor, please add some more details.

Response: Thanks for pointing this out. On lines 186-196 you can see these changes:

“This scale is composed of eight items that assess the frequency of health symptomatology. Specifically, individuals were asked to respond how often they have felt a list of symptoms (e.g., anxiety, depression, stress, sleeping problems, eating problems, stomach problems, sexual problems and headaches) using a 7-point Likert-type scale (1 = never to 7 = very often). Higher scores reflected a greater presence of this type of symptomatology. A previous study (31) revealed the presence of two factors, one grouping the first three items in mental health symptoms and the rest in physical symptoms. Also showed that the correlation between the mental health symptoms factor and physical health symptoms was high. In this study, the reliability for the mental health and physical factors was good, with alpha coefficients of .746 and .701, respectively”.

Comments: 4.- Regarding the sample, why was the final sample size considered in both studies? How was the estimation made?

Response: Thanks for pointing this out. We have included on lines 269-272 how we estimated sample size:

“The sample size was estimated based on the absolute number of cases criterion. Specifically, the estimation was calculated with a 95% confidence level and a 5% margin of error of the total number of immigrants living in the La Araucanía region”.

Comments: 5.- In the sample of study 1, university students in one age range were considered, however, in study 2 a wide age range was considered, why was the age range widened in study 2?

Response: The main objective of Study 2 was to assess the immigrant population residing in a specific region in the south of Chile. Given that this is a new area of migration, the selection criteria for participants were expanded in order to meet the estimated sample size. The only criterion for participants was to be older than 18.

Comments: 6.- DASS-21 has been used to correlate with the mental and physical health symptomatology scale, in this case because a scale with dimensions of physical functioning has not been used.

Response: As stated in the background, physical symptomatology has received less attention in research, and there are few instruments available to assess this dimension. Although the DASS-21 primarily assesses mental health, it also captures some associated physical symptoms, recognizing the interrelatedness of physical and mental symptomatology. This study hypothesized a high correlation between the DASS-21 and the mental health factor of the new scale. Future studies could further explore this relationship and investigate how the MPHS-Scale correlates with a physical health instrument, as discussed on lines 447-461. We hope this addresses the comment appropriately.